# Group B streptococcal colonization: Prevalence and impact of smoking in women delivering term or near term neonates in a large tertiary care hospital in the southern United States

**Philip Kum-Nji**[1]*, **Linda Meloy**[1], **John Pierce**[2], **Amanda Ritter**[3], **Rachel Wheeler**[3]

1 Children's Hospital of Richmond at the Virginia Commonwealth University School of Medicine, Richmond, Virginia, United States of America, 2 Lynchburg Women's Health, Lynchburg, Virginia, United States of America, 3 Department of Obstetrics and Gynecology, Virginia Commonwealth University School of Medicine, Richmond, Virginia, United States of America

* pkumnji@vcu.edu

**Data Availability Statement:** All relevant data are within the manuscript and its Supporting Information files.

## Abstract

### Background and hypothesis

The role of smoking as a risk factor for group B streptococcal (GBS) colonization in women during pregnancy has not been previously adequately explored. We hypothesized that women of term or near term neonates who smoked during pregnancy were more likely to have GBS colonization than their non-smoking counterparts.

### Methods

The electronic health records (EHRs) of a convenience sample of women delivering in an inner-city university tertiary care center were reviewed. The outcome variable of interest was maternal GBS colonization during pregnancy. The primary independent variable of interest was tobacco smoking during pregnancy, determined from the EHRs by the number of cigarettes smoked during gestation. Descriptive statistics were conducted and categorical data were compared by the Fischer's exact test. Multiple logistic regression analysis was further conducted to determine the independent impact of tobacco smoke exposure on GBS colonization.

### Results

The prevalence of maternal GBS colonization was 35% among the study population. In the univariate analyses, factors associated with maternal GBS colonization were tobacco smoking during pregnancy (P of trend <0.001), Race (P<0.001), maternal age <20 years (P = 0.006), low birthweight <2500 gm (P = 0.020), maternal drug use (P = 004), and gestational age <37 (P = 0.041). In a multiple logistic regression analysis, tobacco smoking during pregnancy remained the most significant predictor of GBS colonization. Women who smoked during pregnancy were more than twice more likely to be colonized than their non-smoking counterparts (OR = 2.6; 95% CI = 1.5–4.6; p<0.001). Maternal age was the only other

**Funding:** The authors received no specific funding for this work.

**Competing interests:** The authors have declared that no competing interests exist.

significant predictor with younger mothers more than one and a half time more likely to be colonized than their older counterparts (OR = 1.65; 95% CI = 1.02–2.68; P = 0.04).

## Conclusion

The prevalence of GBS colonization in this institution was consistent with recent national rates. Smoking and maternal age were identified as two independent risk factors for GBS colonization during pregnancy. Further studies are needed to confirm these findings.

## Introduction

Maternal group B streptococcal (GBS) colonization is one of the most important risk factors for GBS disease in neonates [1]. In 2002, the Centers for Disease Control (CDC) recommended universal culture-based screening of all pregnant women between 35–37 weeks gestation in order to treat all those positive for GBS with intrapartum antibiotics prior to delivery to prevent against early onset neonatal GBS sepsis [2]. Ever since the institution of this policy, the incidence of early onset GBS sepsis has continued to fall dramatically from a rate of 1.5/1,000 to a rate of 0.24/1,000 live births [3].

The GBS colonization rates vary from country to country, the geographic location, and the method used in determining the colonization. In the USA, since the early 80's colonization rates ranged from 20% to 35% [4–6]. In Africa, rates have ranged from 20% to 30% [7–9]. In Eastern Europe rates of 30+% have also been determined [10]. In Western Europe rates have ranged from a low of 14% to a high of 36% [6, 11]. In India one study found a rate of 15% [12]. while in Korea, the rate was as low as 8% [13]. In Australia, rates as high as 35% have been demonstrated [14]. In summary, rates of GBS colonization have remained remarkably stable over the past 40 years worldwide.

Tobacco smoke exposure during pregnancy is one risk factor for GBS colonization that has not yet been adequately explored. Tobacco smoke has been shown to be associated with increased colonization of the respiratory, gastrointestinal, and even genital tracts with various bacterial pathogens [15–21]. However, the impact of tobacco smoke exposure on GBS colonization has not been adequately explored. Because over 10% of women in the United States smoke during pregnancy [22], it is important to clearly determine its impact on GBS colonization. We therefore hypothesized that tobacco smoking during pregnancy is independently associated with increased GBS colonization among women delivering term or near term neonates in this large tertiary care hospital in Virginia.

## Methods

### Ethics statement

The Institutional Review Board (IRB) of the Virginia Commonwealth University School of Medicine approved the study as expedited since the data were purely retrospectively collected from the EHRs. There was no interaction between the subjects and the researchers in any way, hence the need for consent was waived.

### Design and setting

This was a retrospective cross-sectional study design. The study was carried out in a large inner-city tertiary care university teaching hospital in Virginia. Almost 2,000 babies are born

each year in this hospital of which about 30% are Whites, 40% African Americans and 30% Hispanics or Latinos.

## Participants/subjects

Mother-newborn dyad singletons were retrospectively recruited in the study using electronic health records (EHRs). The study was initiated in January 1, 2011 and ended in December 31, 2019. Each year subjects were randomly selected by 2 of the study authors (PK and RW). The authors, at their convenience, were allowed by the IRB to fully access the EHRs of mothers and their neonates anytime during the above study period. Inclusion criteria consisted of only women delivering term or near-term neonates 35+ weeks gestational age. Women of preterm neonates < 35 weeks were excluded from the study.

## Variables and definitions of terms

The outcome variable of interest was GBS colonization of pregnant mothers during gestation. GBS colonization status was determined by recto-vaginal swabs at 35–37 weeks gestation and consisted of placement of the genital swab approximately 2 cm into the vagina followed by placement through the anal sphincter. The result of the test was defined as either positive or negative by the PCR using standard techniques as described [23]. Tobacco smoking was defined by the number of cigarettes smoked per day during gestation. This variable was accurately recorded by the attending obstetrician during prenatal visits. In the rare instances when this information was missing in the EHR of the obstetrician, e.g. because of lack of prenatal care, the pediatrician usually recorded the smoking status after delivery by direct questioning of the new mother during her Nursery stay. If the mother smoked during pregnancy, the number of cigarettes was also recorded in the mother's EHR by the obstetrician or the pediatrician. Information on tobacco smoking status of all subjects was thus available. Maternal variables abstracted were: age in years, parity, gestational diabetes, race, and substance use. Infant factors abstracted were: birthweight in grams, gestational age in weeks, and gender of the newborn baby (male or female). Maternal substance use was defined as any use during pregnancy of various drugs such as cocaine, various opiates (e.g., methadone, OxyContin, buprenorphine (Subutex), cannabinoids, or other illegal substances. These potential risk factors were all gleaned from the literature.

## Data analyses

A total sample of 736 mother newborn dyad was recruited in the study. The outcome variable of interest was GBS colonization among pregnant women delivering term or near term infants 35+ weeks gestational age. The primary independent variable of interest was active tobacco smoking throughout pregnancy as determined by the number of cigarettes smoked per day. In the univariate analysis, tobacco smoking was categorized as 0 cigarette/day, 1 cigarette/day, 2 cigarettes/day, and 3 + cigarettes/day respectively for trend analysis. Test of trend was also explored for maternal age. Initially maternal age was empirically categorized into 2 groups (< 20 year of years versus 20+ years). A trend analysis for this variable was then conducted to determine whether GBS rates varied by the various maternal age groups. When a clear trend emerged, maternal age was subsequently empirically expanded to 5 groups (<20, 20–24, 25–29, 30–34, and 35+) to explore whether the trend continued. Tests of trends were also explored for the other independent variables of interest (birthweight, gestational age) but were not significant so these variables were left dichotomized in the univariate analysis. In the multiple logistic regression analyses, all the independent variables of interest were left dichotomized for simplicity of analyses.

Descriptive statistics were conducted. Univariate analysis was conducted to determine if each of the selected variables was associated with maternal GBS colonization using Fischer's exact test for comparisons of proportions. Next, a multiple logistic regression analysis was conducted using only the statistically significant variables in the univariate analysis to determine if each of these factors was still predictive of maternal GBS colonization after controlling for the other potential confounders. The SPSS statistical software (IBM, Armonk, NY 2018) was used in the analyses. A p value of <0.05 was used as test of significance in all cases.

## Results

### Characteristics of the study population

In this study almost 35% of the mothers were colonized with GBS (see Table 1). Overall 11% of the mothers smoked during pregnancy, 2.6% smoked 1 cigarette per day while almost 9% smoke two or more cigarettes daily. The rest of the Table is self-explanatory.

### Prevalence of GBS colonization by sociodemographic characteristics

GBS colonization was least prevalent among the none smokers (32%) but rose sharply to 50% if the subjects smoked 1 cigarette/day and the rate almost doubled to 61% if the subjects smoked 2+cigarettes/day (p of trend = 0.001). When maternal age was categorized into 2 groups (teens vs older), GBS was higher among the teen mothers (<20 years) as compared to their older counterparts (20+ years, P = 0.004; also see Table 2). Furthermore, when maternal age was categorized into 5 groups, the prevalence of GBS was highest among women <20 years (49%), dropped to 40% in women 20–24 years, and 30% in women 25 to 29years. However, there was a small spike to 36% in women 30–35 years but the rate dropped again to 28% in women 35 + years. In general, therefore, there was a decreasing trend of GBS colonization with increasing maternal age (P of Trend = 0.023; also see Fig 1). Interestingly, this trend was only true among non-smokers. Among the smokers, however, there was no downward trend of maternal in GBS colonization by maternal age (P of Trend = 0.935; also see Fig 2).

GBS colonization varied widely among the racial groups and was lowest among Asian /Pacific Island Americans (22.5%) as compared to the Latinos (26%), White Americans (30.6%) or African American (49%, P <0.001). Furthermore, the proportion of colonization was more prevalent among women with any history of substance use (54%) as compared to those without any use (35%; P = 0.004). Thus, of the nine variables explored in this study, only 3 viz. gestational diabetes mellitus, parity, and gender were not predictive of GBS carriage (also see Table 2).

### Multiple logistic regression of factors associated with GBS colonization

All the 6 variables significantly associated with GBS colonization in the univariate analyses (smoking, maternal age, birthweight, gestational age, race, and use of substance) were now selected for inclusion in the multiple logistic regression analysis. Tobacco smoking was by far the most significant predictor of GBS colonization even after controlling for the other confounders (adjusted OR = 2.6, 95% CI: 1.53–4.34; P <0.001). The only other risk factor independently associated with GBS colonization was maternal age < 20 years (adjusted OR = 1.65, 95% CI: 1.02–2.68; P = 0.041, also see Table 3).

## Discussion

There were three important findings in this study: 1) Almost a third of all women delivering term or near neonates were colonized with GBS in this hospital; 2) even after controlling for

**Table 1. Sociodemographic characteristics of the study population.**

| Continuous Variables N = 736) | Mean (SD) | Range |
|---|---|---|
| Maternal Age (years) | 28.1 (5.6) | 15–44 |
| Gestational Age (weeks) | 39.1 (1.4) | 35–42 |
| Birthweight (grams) | 3335.2 (485.7) | 2030–4805 |
| Parity | 2.2 (1.3) | 0–9 |
| Number of cigarettes/day | 0.6 (2.5) | 0–30 |
| Categorical Variables | Number (%) | |
| Group B Strep Colonized? | | |
| Yes | 257 (34.9.) | |
| No | 479 (65.1) | |
| Mother Smoked? | | |
| Yes | 84(11.4.0) | |
| No | 652 (88.6) | |
| Maternal Age (years) | | |
| < 20 | 84 (11.4) | |
| 20+ | 652 (88.6) | |
| Birthweight (grams) | | |
| <2500 | 27 (3.7) | |
| 2500 + | 709(96.3) | |
| Gestational Age (weeks) | | |
| < 37 | 32 (4.3) | |
| 37+ | 704 (96.7) | |
| No of Cigarettes per day | | |
| 0 | 652 (88.6) | |
| 1 | 19 (2.6) | |
| 2 | 19 (2.6) | |
| 3+ | 46 (6.2) | |
| Gender | | |
| Males | 365 (49.6) | |
| Females | 371 (50.4) | |
| Race/Ethnicity | | |
| White | 242 (32.9) | |
| African American | 249 (33.9) | |
| Hispanic/Latino | 174 (23.6) | |
| Asia/Pacific Islander | 71 (9.6) | |
| Maternal Substance Use | | |
| Yes | 46(6.3) | |
| No | 690 (93.7) | |
| Infant of Diabetic Mother | | |
| No | 669 (91.8) | |
| Yes | 60 (8.2) | |
| Missing | 7 (1.0) | |
| Parity | | |
| <2 | 269 (36.5) | |
| 2+ | 467 (63.5) | |

various confounders, smoking during pregnancy was more than twice more likely to result in GBS colonization among pregnant women of term or near term neonates.; 3) teen mothers less than 20 years had the highest rates of GBS colonization and the rates steadily decreased

**Table 2. Prevalence of GBS colonization by sociodemographic characteristics.**

| Variable (N = 736) | Number (% with MGBS) | P Value |
|---|---|---|
| **Maternal Variables** | | |
| Mother Smoked? | | |
| Yes | 49 (58.3) | |
| No | 208 (31.9) | < 0.001 |
| Tobacco Smoking (No of cigs/day) | | |
| 0 | 210 (32.0) | |
| 1 | 8 (50.0) | |
| 2+ | 39 (60.9) | <0.001 |
| Race | | |
| White | 74 (30.6) | |
| African American | 122 (49.0) | |
| Hispanic | 45 (25.9) | |
| Asian /Pacific Islanders | 16 (22.5) | <0.001 |
| Maternal Age (years) | | |
| < 20 | 39 (48.1) | |
| 20+ | 218 (33.3) | 0.006 |
| Maternal Substance Use | | |
| No | 232 (33.6) | |
| Yes | 25 (54.3) | 0.004 |
| Gestational Diabetes* | | |
| No | 230 (34.4) | |
| Yes | 25 (41.7) | 0.257 |
| Parity | | |
| <2 | 85 (31.6) | |
| 2+ | 172 (36.8) | 0.088 |
| **Infant Variables** | | |
| Gestational Age (Weeks) | | |
| <37 | 241 (34.2) | |
| 37+ | 16 (50.0) | 0.051 |
| Birthweight (gm) | | |
| <2500 (LBW) | 15 (55.6) | |
| 2500+ (NBW) | 242 (34.2) | 0.020 |
| Gender of Newborn | | |
| Male | 134 (36.7) | |
| Female | 123 (33.2) | 0.311 |

* Numbers may not add up due to missing information.

with age with the lowest rates occurring among the oldest mothers 35 year+. Interestingly this inverse relationship was only true for women who were nonsmokers.

This high prevalence of GBS colonization found in our study is consistent with previous studies in the USA and elsewhere [4–12]. The lowest rates were observed mainly in some East Asian nations [6, 12, 13]. The across-country differences may be due to differences in techniques [23–35]. PCR testing after broth enrichment has been has been shown to be a reliable and is now the standard method for GBS testing [24]. However, testing conducted intrapartum may show somewhat lower prevalence rates as compared to antepartum testing [25]. If this is so, then there may be excessive use of intrapartum prophylactic antibiotics to prevent GBS

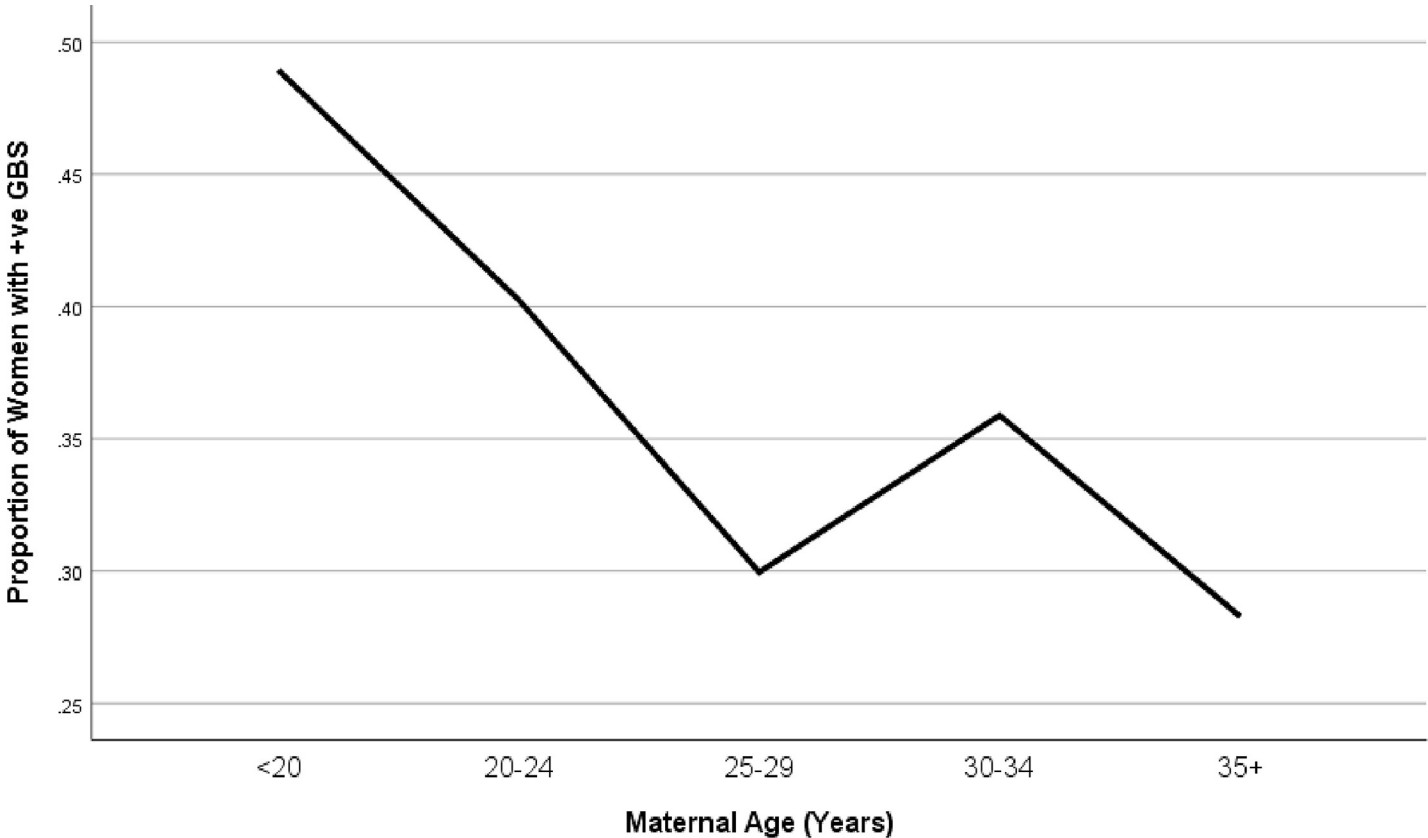

**Fig 1. Prevalence of maternal GBS colonization by age group.** The Figure demonstrates that overall the youngest mothers (<20 years) had the highest rates of GBS colonization (49%) while the oldest mothers (35+ years) had the lowest rates (28%) although there was a small spike in mothers 30 to 34 years (P of Trend = 0.023).

sepsis in neonates. Unfortunately, intrapartum testing is not yet practical since it can take up to 2 to 3 days for the results of tests to be made available whereas knowing the results prior to delivery is very important to make a decision to treat prophylactically or not. Hopefully, new rapid PCR tests are being developed for such a purpose [25]. In spite of the dramatic drop of early onset neonatal GBS sepsis in the United States, the reasons for the persisting high prevalence rates of GBS colonization during pregnancy throughout the past 4 decades need to be further explored.

This study further confirms that, potentially, more than a third of all women were eligible to receive intrapartum prophylactic antibiotics prior to delivery. This is indeed a high rate of pre-delivery antibiotics exposure of term or near term neonates. Several authors have recently suggested that at least 40% of children may be exposed to intrapartum antibiotics [26, 27]. Although intrapartum antibiotic prophylaxis have successfully reduced the incidence of GBS sepsis in neonates, the dangers of such prophylaxis are concerning [28]. For instance, intrapartum antibiotic prophylaxis have been associated with complications such as increased prevalence of atopic dermatitis [29], antimicrobial resistance [30], and changes in the neonatal gut microbiota [21, 31, 32].

The prevalence of active tobacco smoking among pregnant women in this study was 11%, and this is consistent with recent national trends [22]. However, in our study, the intensity of smoking during pregnancy was rather low and most women who smoked admitted to smoking no more than 3 cigarettes each day although this may have been largely an underestimate. Since this was a retrospective study the assumption is that women who admitted to smoking

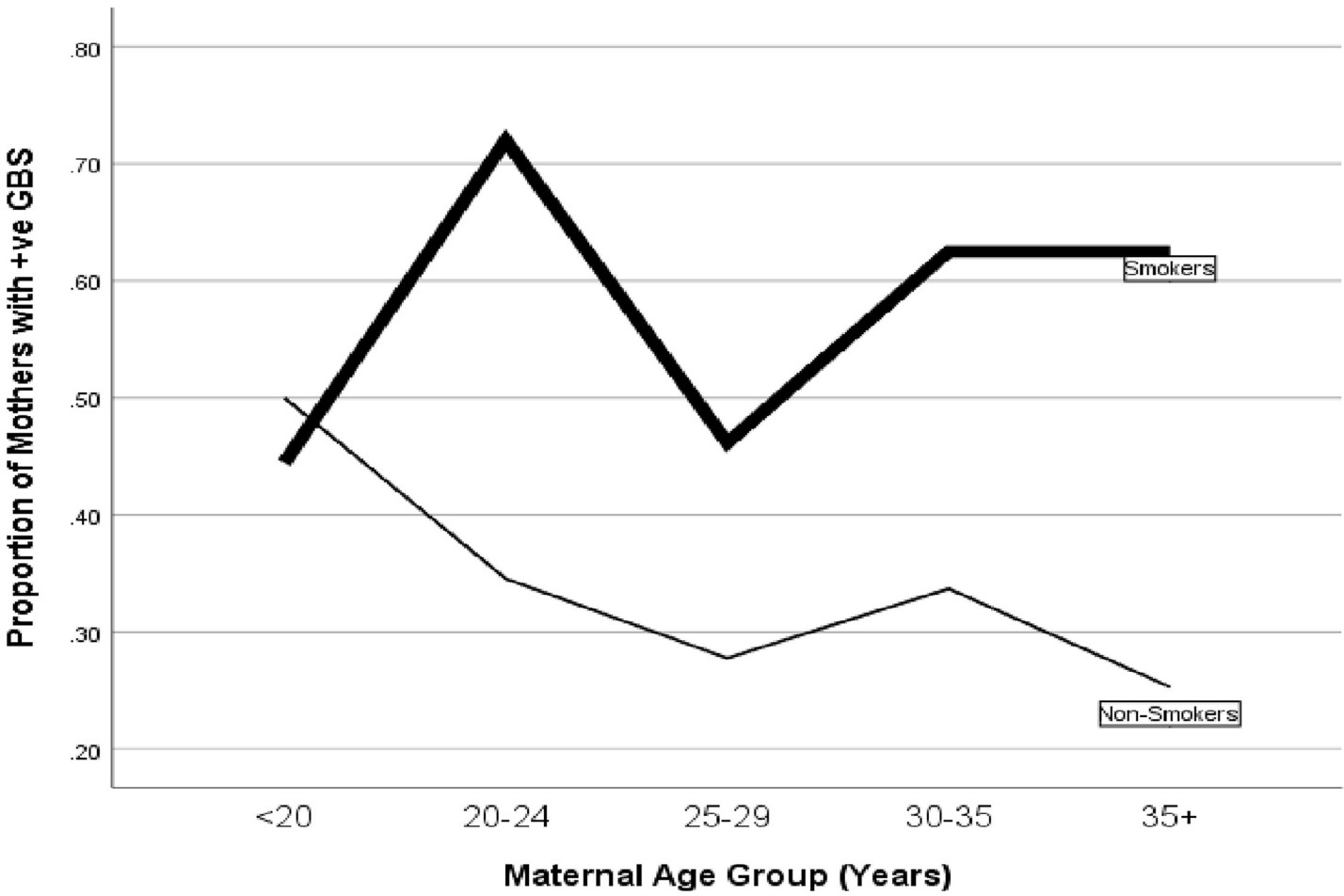

**Fig 2. Prevalence of maternal GBS colonization by maternal age group and smoking status.** This Figure shows that, for smoking mothers, there was no relationship between maternal age and GBS colonization (P of Trend = 0.935) whereas for the non-smoking mothers there was a clear inverse-relationship between maternal age and GBS colonization (P of Trend <0.05).

actually smoked throughout pregnancy. The study may, therefore, not take account of those who stopped smoking sometime during pregnancy or those who only initiated smoking sometime during pregnancy.

Few studies in the literature have explored the impact of smoking during pregnancy on GBS colonization. Our study was also able to demonstrate a significant dose-response relationship suggesting that this was not just a spurious finding. In the United States, Terry et al. [33] were the first to show that smoking during pregnancy was predictive of GBS colonization but the authors did not perform a multivariable analysis to control for other confounders to determine if smoking was independently predictive. Edwards and colleagues [34], in a large retrospective study, recently demonstrated that smoking was also predictive of GBS colonization in the univariate analysis but not in the multiple regression analysis. However, smoking was not the primary independent variable of their study. Two studies respectively from Korea and China [13, 35], failed to demonstrate an association of smoking during pregnancy with GBS colonization. However smoking was also not the primary independent variable of interest of these studies and there were was the additional problem of smallness of numbers in some cells. Our finding that smoking during pregnancy was predictive of GBS colonization during pregnancy is consistent with a smaller study from Iran [36]. However, their study population was

**Table 3. Multiple logistic regression of factors associated with GBS colonization among pregnant mothers showing unadjusted and adjusted Odds Ratios (OR) and their 95% Confidence Intervals (CI).**

| Variables | Unadjusted OR (95% CI) | P Value | Adjusted OR (95% CI) | P Value |
|---|---|---|---|---|
| Mother Smoked? | | | | |
| No (Referent) | 1 | - | 1 | - |
| Yes | 2.99 (1.88–4.75) | <0.001 | 2.55 (1.54–4.34) | <0.001 |
| Maternal Age (years) | | | | |
| 20+(Referent) | 1 | - | 1 | - |
| <20 | 1.86 (1.17–2.96) | 0.009 | 1.65 (1.02–2.68) | 0.04 |
| Birthweight (grams) | | | | |
| 2500+ (Referent) | 1 | - | 1 | - |
| < 2500 | 2.41 (1.11–5.23) | 0.026 | 1.89 (0.83–4.25) | 0.132 |
| Race | | | | |
| White (Referent) | 1 | - | 1 | - |
| Non-White | 1.34 (1.0–1.86) | 0.05 | 1.22 (0.87–1.72) | 0.246 |
| Gestational Age (weeks) | | | | |
| 37 + (Referent) | 1 | - | 1 | - |
| <37 | 1.92 (0.94–3.91) | 0.07 | 1.29 (0.83–1.99) | 0.258 |
| Maternal Substance Use | | | | |
| No (Referent) | 1 | - | 1 | - |
| Yes | 2.35 (1.29–4.29) | 0.005 | 1.27 (0.64–2.52) | 0.487 |

different as it also included preterm neonates that were excluded in our current study. Surprisingly, Regan and colleagues demonstrated in 1991 that smoking was rather protective of GBS colonization [37]. The authors offered no biological plausibility of their finding. However, only women delivering preterm neonates (<36 weeks) were enrolled in the latter study whereas our study focused only on term or near term neonates. We speculate that the population of women delivering term babies may be quite different from the population delivering only preterm babies.

The finding that tobacco smoke exposure during pregnancy is an independent risk factor for increased GBS colonization has significant public health implications. Because smoking exposure is a modifiable risk factor, women can be counseled to stop smoking during pregnancy in order to reduce colonization with this organism which can result in GBS sepsis in their newborn baby. The association between tobacco smoking and increased GBS colonization during pregnancy is biologically plausible. We speculate that tobacco smoke exposure during pregnancy may actually enhance the colonization of GBS in the gastrointestinal and the genital tracts. Indeed, previous studies have shown that tobacco smoke exposure is associated with increased colonization of the respiratory and genital tract with pathogenic bacteria [38, 39], possibly through alteration of the microbiome [18, 21, 40]. Tobacco smoke contains more than 4,500 chemical intoxicants [41] many of which can result in increased suppression or modulation of both active and passive immune response [42]. For instance, nicotine, an important component of tobacco smoke, has been shown to enhance the adherence of bacteria in mucous membranes of the respiratory tract leading to easy penetration of bacteria into the tissues to cause infections [43]. This could also be true of the genital tract of pregnant women where nicotine could actually result in the persistence of GBS in the mucous membranes of the gastrointestinal and genital tracts. Indeed some studies have actually demonstrated higher nicotine levels in the cervical mucous membranes of smokers as compared to non-smokers [44, 45]. In another study, nicotine of the cervical mucus of female smokers resulted in DNA

damage of epithelial cells of these women resulting in easy penetration of bacteria into the adjacent tissues [46]. It can be speculated that the overall effect of tobacco smoke is the alteration of the microbiome of the female genital tract resulting in increased prevalence of pathogenic microorganisms such as GBS in the present study.

To our knowledge only a few studies in the United States have been conducted in recent years to determine potential sociodemographic risk factors associated with this rather high colonization rates among pregnant women. In the '70's Anthony et al showed that Mexican Americans had the lowest rates as compared to Whites or Blacks [47]. In our study, black mothers had the highest rates as compared to White or Hispanic/Latino mothers. These findings are consistent with those of Regan and colleagues whose study was conducted almost 30 years ago [37]. The finding that younger mothers had significantly higher rates of GBS colonization was consistent with one previous study conducted more than 40 years ago by Anthony and colleagues [47]. They demonstrated that younger mothers had higher rates of GBS colonization than their older counterparts. Indeed, GBS sepsis in neonates has also been shown to be more common among young mothers < 20 years, as demonstrated by Schuchat et al. [48, 49]. This may be explained by the fact that young mothers also have higher rates of GBS colonization than their older counterparts as shown in our study. However, 10 years after the study by Anthony et al [47], Regan et al [37], showed that GBS colonization was less common among women less than 20 years of age even after controlling for the other sociodemographic confounders. The difference in findings between the two studies may be due to the fact that cultures for GBS were obtained very early in gestation (23–26 weeks gestation) in the latter study, whereas the current recommendation is to obtain cultures at 35–37 weeks of gestation [2]. Again as stated above, our study clearly demonstrates that maternal age was only predictive of GBS colonization among the non-smoking women. Non-smokers < 20 years had the highest GBS colonization rate while those >35 years of age had the lowest rate.

This study has several limitations. First the retrospective nature of the data implies that there is no causality attributed to the findings. Second, because the study only involved subjects recruited from one local hospital, there may therefore be lack of both internal and external validity of the findings as this sample was not necessarily representative of the population of the state or of the nation. Third, the information of tobacco smoking was retrospectively obtained so there is likelihood of misclassification bias. The association between tobacco smoking and GBS colonization would have been even stronger if the smoking status was determined objectively by the use of a biomarker such as serum or urine cotinine levels. This may have reduced the likelihood of misclassification bias of the smoking status of the subjects. However, some of the findings are consistent with previous works. Fourthly, we were not able to control for all the confounders such as obesity and the frequency of sexual relationships which have been shown in some studies to be predictive of GBS colonization [49]. The significance of this study lies in the fact that this is the first robust study with the main focus of determining the impact of tobacco smoke on GBS colonization on women of term or near neonates. In other studies GBS was not the main focus and a test of trend was not explored.

## Summary and conclusion

This is one of the few studies to clearly determine that smoking during pregnancy is an independent predictor of GBS colonization specifically in women of term or near term neonates. GBS colonization was highest among the youngest mothers and tended to decrease with age especially among non-smoking mothers. This is another reason why women should be counseled to stop smoking if they are pregnant in order to avoid risking their newborn from developing GBS disease.

## Supporting information

**S1 Dataset.**
(SAV)

## Acknowledgments

We wish to thank the staff of the Department of General Pediatrics of the Children's Hospital of Richmond for their help in bringing this work to fruition.

## Author Contributions

**Conceptualization:** Philip Kum-Nji, John Pierce.

**Data curation:** Philip Kum-Nji, Linda Meloy, John Pierce, Amanda Ritter, Rachel Wheeler.

**Formal analysis:** Philip Kum-Nji, Linda Meloy, John Pierce, Amanda Ritter, Rachel Wheeler.

**Investigation:** Philip Kum-Nji, Linda Meloy.

**Methodology:** Philip Kum-Nji, Linda Meloy, John Pierce, Amanda Ritter.

**Validation:** John Pierce, Rachel Wheeler.

**Writing – original draft:** Philip Kum-Nji, John Pierce, Amanda Ritter, Rachel Wheeler.

**Writing – review & editing:** Philip Kum-Nji, Amanda Ritter.

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
