## [Decision Letter · Decision Letter 0]

12 Jun 2020

PONE-D-20-15744

GBS colonization: prevalence and impact of smoking in women delivering term or near term neonates in a large tertiary care hospital in the southern United States

PLOS ONE

Dear Dr. Kum-Nji,

Thank you for submitting your manuscript to PLOS ONE. After careful consideration, we feel that it has merit but does not fully meet PLOS ONE’s publication criteria as it currently stands. Therefore, we invite you to submit a revised version of the manuscript that addresses the points raised during the review process.

SPECIFIC ACADEMIC EDITOR COMMENTS: An expert reviewer in the field handled your manuscript. Limited interest was found in your manuscript, but the reviewer offers suggestions to strengthen the merit of this study. These suggestions relate to the need to complete the STROBE checklist; the introduction needs to be rewritten to strengthen the rationale and readability; clarification of the statistical analyses is warranted; the results section needs to be more succinct; and a better conclusions statement should to be provided.

We look forward to receiving your revised manuscript.

Kind regards,

Frank T. Spradley

Academic Editor

PLOS ONE

2. Please provide additional details regarding participant consent. In the ethics statement in the Methods and online submission information, please ensure that you have specified (1) whether consent was obtained, (2) whether consent was informed, and (3) what type you obtained (for instance, written or verbal, and if verbal, how it was documented and witnessed). If your study included minors, state whether you obtained consent from parents or guardians. If the need for consent was waived by the ethics committee, please include this information.

3. Please provide the dates that you accessed the data in your retrospective study.

4. Please include a caption for each figure.

5. Please include your tables as part of your main manuscript and remove the individual files. Please note that supplementary tables should remain as separate "supporting information" files.

6. We note you have included a table to which you do not refer in the text of your manuscript. Please ensure that you refer to Table 3 in your text; if accepted, production will need this reference to link the reader to the Table.

Reviewers' comments:

Reviewer's Responses to Questions

**Comments to the Author**

1. Is the manuscript technically sound, and do the data support the conclusions?

Reviewer #1: Partly

2. Has the statistical analysis been performed appropriately and rigorously? 

Reviewer #1: Yes

3. Have the authors made all data underlying the findings in their manuscript fully available?

Reviewer #1: Yes

4. Is the manuscript presented in an intelligible fashion and written in standard English?

Reviewer #1: Yes

5. Review Comments to the Author

Reviewer #1: PONE-D-20-15744

GBS colonization: prevalence and impact of smoking in women delivering term or near term neonates in a large tertiary care hospital in the southern United States

This manuscript describes the relationship between smoking during pregnancy with group B streptococcal (GBS) colonization among women of term and near term neonates. While the manuscript is of some interest, I do not think the manuscript in its current form is of sufficient interest and quality to warrant publication in PLOS ONE.

I suggest the authors to look for STROBE check list for cross-sectional studies to ensure reporting is complete and transparent.

https://www.strobe-statement.org/fileadmin/Strobe/uploads/checklists/STROBE_checklist_v4_cross-sectional.pdf

My sense is that the manuscript could be strengthened by several modest changes as outlined below. Inserting line numbers may facilitate to give comments and feedbacks.

Title

- Suggest to expand GBS long form. Otherwise, the readers may confuse with Genotyping-by-sequencing or Guillian-Barre Syndrome or etc.

INTRODUCTION

- Introduction section is not well organized and not well-written.

- Introduction section needs to include - What is the problem and its supporting details? What is the added scientific value from the study and why are you conducting it? - Both globally and locally.

- Paragraph 2 – the information is overloaded and recommended a concise statement. Some detailed information can be shown by references.

- Need to add reference for the sentence “Maternal group B streptococcal (GBS) colonization is one of the most important risk factors for GBS disease in neonates.”

- The authors described “Potential sociodemographic risk factors for GBS colonization among pregnant women have not been recently explored. Perusal of the literature within the last 2 decades demonstrates a paucity of information on other potential sociodemographic risk factors associated with maternal GBS colonization during pregnancy.” in the introduction but I find myself many papers in the references as well as online. Moreover, this does not reflect the justification of the stated hypothesis.

METHODS

- The authors mentioned “the data ….. divided almost equally yearly (i.e. almost 90 patients selected each year)”. Is this selected by the authors or by the data set?

- Abbreviate first before using it (E.g. IRB, x2)

- Check grammar and spelling mistake throughout the manuscript

- The authors mentioned only the significant variables were put for a multiple logistic regression analysis. What do authors mean by significant? Statistically significant? What is the cut-off point, <0.05 or <0.2?

- Many items of the STROBE check list are written; however, most of them are jumbled up under different sub heading and some are missing (inclusion criteria, exclusion criteria, etc.)

RESULTS

- The authors tend to cover all information in the text and again in the table. You should only include the most important details in the text when you also have a table.

- Any reason behind for including 67 women who outcome status were unknown as study population?

- Any reason behind for categorizing maternal age in two ways?

- Trend analysis, Figure 1 and 2 may not be necessary for testing the study hypothesis.

- Foot note is missing for * in table 2

- Tables should be structured properly (check the published PLOS ONE papers).

- To me, this is a cohort study which the authors can calculate and present relative risk (strong statistic).

- All things considered, re-analysis and rewriting of results section is necessary.

DISCUSSION

- The discussion is the best part of the manuscript.

- Repeating the finding statements and analysis result (E.g. P of trend = 0.02) should be avoided here.

CONCLUSION

- The authors concluded “GBS colonization as rather high but consistent with recent national trend” which is not clear.

6. PLOS authors have the option to publish the peer review history of their article (what does this mean?). If published, this will include your full peer review and any attached files.

Reviewer #1: Yes: Kyaw Lwin Show

---

## [Author Response · Author response to Decision Letter 0]

16 Jul 2020

RESPONSES TO REVIEWER of PLoS ONE: Group B Streptococcal colonization: prevalence and impact of smoking in women delivering term or near term neonates in a large tertiary care hospital in the southern United States

The Editor PLoS ONE,

Thank you for allowing us to rewrite our manuscript titled above following your recommendations and those of the Reviewer.

RESPONSES TO SPECIFIC ACADEMIC EDITOR COMMENTS

1) The PLoS ONE’s style requirements were followed

2) Consent Issues: As stated in out manuscript (page line ) the manuscript was reviewed as expedited and no consent was required since this was purely retrospective data from the HER. There was no interaction between the subjects and the researchers in any way, hence the need for consent was waived by the IRB of the Virginia Commonwealth University.

3) EHRs were assessed variously on throughout May 1 to July 31 of 2011, and subsequently same dates through 2019

4) A caption of Fig 1 and Fig 2 included

5) We have included the Tables as part of the main manuscript

6) Table 3 has been referenced in the main manuscript. Thanks for spotting that error

7) Captions for the Supporting Information files at the end of the manuscript have now been included.

RESPONSES TO REVIEWER’s COMMENTS

We have endeavored to follow the STROBE checklist while rewriting this manuscript as suggested by the Reviewer # 1. We furthermore inserted line numbers to facilitate comments and feedback

TITLE

We have changed the abbreviation “GBS” to its expanded form “Group B Streptococcal”. We thank the Reviewer for pointing out this error. 

INRODUCTION

- We have reorganized the Introduction: In paragraphs 1 and 2 we have restated the importance of GBS colonization in causing GBS disease worldwide. In paragraph 3 we have stated that one of the risk factors for GBS colonization not adequately explored is tobacco smoke exposure during pregnancy. If this is found to be an independent risk factor, then this may be one additional reason why women may benefit from refraining from smoking during pregnancy

- Paragraph 1. We have added a very important reference to the opening statement “Maternal group B streptococcal (GBS) colonization is one of the most important risk factors for GBS disease in neonates” (Verani et al). We thank the Reviewer for pointing this out. 

- Paragraph 2. We have deleted the first sentence in this paragraph but we have left the rest of the 2nd paragraph 2 intact even though there may some information overload as stated by this Reviewer.

- As suggested by the Reviewer we have remove a controversial statement about lack of studies on sociodemographic risk factors associated with GBS colonization. We have rewritten this 3rd paragraph to emphasize ONLY the lack of studies on tobacco smoking during pregnancy as a risk factor for GBS colonization. We thank the Reviewer for pointing this out. 

METHODS

- Yes this was a convenience sample selected by the authors form EHRs between May and July of each year.

- Abbreviation issues corrected

- We have endeavored to the best of our ability to check the grammar and spellings as suggested by the Reviewer.

- We have added the word “statistically” in the sentence since only statistically significant variables in the univariate analysis (p<0.05) were included in the multivariate analysis (This is stated see last sentence last paragraph of Data Analysis).

- STROBE checklist: We have endeavored to rewrite and include all items of the checklist although some of the items may not necessarily be in the same order as in the STROBE checklist. We have clearly stated that only women of term or near term neonates were included (35+ week gestational age). All preterm neonates <35 weeks gestational age were excluded.

RESULTS

- We agree with the Reviewer that the Results section may be overloaded with information already in the Tables. We have therefore rewritten the Results section summarizing only the most important findings.

- The Reviewer suggested that we remove the 67 subjects with unknown GBS status in the analyses. This has been done and the sample size is now 736 subjects. The analysis has been redone but mainly affects Table 1 as can be seen. Table 2 and 3 remain unchanged since these tables only took account of the 736 subjects with known GBS status.

- The Reviewer wanted to know why we categorized maternal age (and tobacco smoke) in two different ways. As stated in our Methods section we did a trend analysis only of all the statistically significant independent predictors of GBS colonization in the univariate analysis. In the multiple logistic regression analysis, all variables were dichotomized for simplicity. 

- The Reviewer felt trend analyses and Fig 1 and 2 may not be needed. However we feel that a trend analysis gives the average reader additional insight on how smoking and maternal age affect GBS colonization. The Figures clearly show that maternal age is only predictive of GBS among the non-smoking women regardless of age group. 

- Footnote of Table 2 has been inserted

- The Reviewer suggests that this study should be labeled as a cohort. But because all the data were retrospectively collected from EHRs without any interactions with the subjects this is still a retrospective study and we still think Odds Ratios rather than relative risks should be calculated.

DISCUSSION

- We are very thankful for this Reviewer’s appreciation of our discussion in this study.

- We have followed the Reviewer’s suggestion to avoid any finding statements and analysis result have been deleted these (e.g P of Trend=0.02)

---

## [Decision Letter · Decision Letter 1]

23 Jul 2020

PONE-D-20-15744R1

Group B streptococcal colonization: prevalence and impact of smoking in women delivering term or near term neonates in a large tertiary care hospital in the southern United States

PLOS ONE

Dear Dr. Kum-Nji,

Thank you for submitting your manuscript to PLOS ONE. After careful consideration, we feel that it has merit but does not fully meet PLOS ONE’s publication criteria as it currently stands. Therefore, we invite you to submit a revised version of the manuscript that addresses the points raised during the review process.

SPECIFIC ACADEMIC EDITOR COMMENTS: There are some additional revisions that are requested and must be addressed.

We look forward to receiving your revised manuscript.

Kind regards,

Frank T. Spradley

Academic Editor

PLOS ONE

Reviewers' comments:

Reviewer's Responses to Questions

**Comments to the Author**

1. If the authors have adequately addressed your comments raised in a previous round of review and you feel that this manuscript is now acceptable for publication, you may indicate that here to bypass the “Comments to the Author” section, enter your conflict of interest statement in the “Confidential to Editor” section, and submit your "Accept" recommendation.

Reviewer #1: All comments have been addressed

2. Is the manuscript technically sound, and do the data support the conclusions?

Reviewer #1: Yes

3. Has the statistical analysis been performed appropriately and rigorously? 

Reviewer #1: Yes

4. Have the authors made all data underlying the findings in their manuscript fully available?

Reviewer #1: Yes

5. Is the manuscript presented in an intelligible fashion and written in standard English?

Reviewer #1: Yes

6. Review Comments to the Author

Reviewer #1: PONE-D-20-15744-R1

Group B Streptococcal colonization: prevalence and impact of smoking in women delivering term or near term neonates in a large tertiary care hospital in the southern United States

My sense is that the manuscript could be strengthened by minor modifications as outlined below.

- Check the spelling mistakes (E.g. LINE 89)

- Table 2 – what do you mean *? In the responses, the authors stated that footnote was inserted but I could not find it.

- Check number in LINE 152, 153 – first 736, out for 67 , then 736 again

- Discrepancies of proportion in the text and table (check carefully)

- The above errors mean that the authors should have checked every single word in the manuscript before submission. There may be more that I couldn’t find.

- The authors stated “Using x2 analysis of comparisons of proportion with Fisher’s exact test” which is not clear. Used chi-square or Fisher’s exact?

- Table 1 - Maternal smoked during pregnancy? � 803 participants?

- Table 3 should be revised (check Table 4 as example from https://journals.plos.org/plosone/article?id=10.1371/journal.pone.0176875)

- The authors concluded “GBS colonization as rather high but consistent with recent national data” which is not clear. Why do you say high here? Is nationally already high? High compared to what?

7. PLOS authors have the option to publish the peer review history of their article (what does this mean?). If published, this will include your full peer review and any attached files.

Reviewer #1: No

---

## [Author Response · Author response to Decision Letter 1]

28 Jul 2020

RESPONSES TO THE ACADEMIC EDITOR AND REVIEWER

We have checked the spelling mistakes and corrected lines 88 and 89: “out” is inserted between “carried” and “in” to read “carried out in” Also in a large “hospital.in Virginia” now becomes “hospital in Virginia”. Thus the period between “hospital” and “in” is deleted

Table 2. We have deleted the asterisk “*” from this Table (see “Tobacco Smoking (No of cigs/day)” and “Race”. Since we have reanalyzed the data there is no longer a need for a footnote for these variables. We thank the editor for consistently spotting this error.

Lines 152-153: We have deleted this sentence completely since it was indeed rather confusing and did not make sense. We thank the reviewer for spotting this error!

Discrepancies of proportions in the text and Tables. We again thank the reviewer for spotting these errors! After reanalyzes of the data as suggested by the previous reviewer we completely forgot to go back and to very carefully edit the numbers in the texts. All these have now been carefully corrected as listed in the lines of the new “Track Changes” as noted below:

Table 1:

- Line 150 : “2%” becomes “2.6” 

- Line 151: becomes “10%” becomes “9%” 

Table 2:

- Line 155: 35% becomes “32%” and “53%” becomes “50%”. 

- Line 156: “63%” becomes “61%” .

- Line 157: “P of Trend= 0.003” becomes “P of Trend=0.001” .

- Line 159: “P=0.006” becomes “P=0.004”.

- Line 169: Pacific Island Americans (22%)” becomes “22.5%”. 

- Lines 169-170: “White Americans (30.1%)” now becomes “(30.6)” 

Data Analysis : Lines 140-141: We have deleted the portion on X2 analyses and left it with Fischer’s exact test since this is what was used in the analyses throughout

Table 1. “Mother Smoke during Pregnancy? “ the numbers have now been corrected and they add up to 736 not 803 

Table 3. This Table has been revised as suggested by the academic reviewer: Unadjusted ORs have therefore been added to the adjusted ORs in the previous analysis and heading of the Table has been updated.

Conclusion: Because there is confusion about the use of the word “high”, the authors have decided to simply delete the use of this word. The conclusion has therefore been slightly reworded. 

We very much appreciate the academic reviewer for their thorough editing of our manuscript and their suggestions for improvement. We hope you can publish it in its present form. 

Please, do not hesitate to let us know if we can further improve our manuscript fit for publication in your esteemed journal. 

Philip Kum-Nji, MD, MPH

(On behalf of the authors)

---

## [Decision Letter · Decision Letter 2]

3 Aug 2020

PONE-D-20-15744R2

Group B streptococcal colonization: prevalence and impact of smoking in women delivering term or near term neonates in a large tertiary care hospital in the southern United States

PLOS ONE

Dear Dr. Kum-Nji,

Thank you for submitting your manuscript to PLOS ONE. After careful consideration, we feel that it has merit but does not fully meet PLOS ONE’s publication criteria as it currently stands. Therefore, we invite you to submit a revised version of the manuscript that addresses the points raised during the review process.

SPECIFIC ACADEMIC EDITOR COMMENTS: There are still comments that must be addressed. Also, please have your manuscript proofed by a professional to assess English grammar and syntax.

We look forward to receiving your revised manuscript.

Kind regards,

Frank T. Spradley

Academic Editor

PLOS ONE

Reviewers' comments:

Reviewer's Responses to Questions

**Comments to the Author**

1. If the authors have adequately addressed your comments raised in a previous round of review and you feel that this manuscript is now acceptable for publication, you may indicate that here to bypass the “Comments to the Author” section, enter your conflict of interest statement in the “Confidential to Editor” section, and submit your "Accept" recommendation.

Reviewer #1: (No Response)

2. Is the manuscript technically sound, and do the data support the conclusions?

Reviewer #1: Yes

3. Has the statistical analysis been performed appropriately and rigorously? 

Reviewer #1: Yes

4. Have the authors made all data underlying the findings in their manuscript fully available?

Reviewer #1: Yes

5. Is the manuscript presented in an intelligible fashion and written in standard English?

Reviewer #1: Yes

6. Review Comments to the Author

Reviewer #1: PONE-D-20-15744-R2

Group B Streptococcal colonization: prevalence and impact of smoking in women delivering term or near term neonates in a large tertiary care hospital in the southern United States

Comments

- I agree about adding unadjusted OR however, Table 3 in current form is UNACCEPTABLE. The readers may get confused regarding the reference category.

- There are typo errors still remained especially in the tables. The authors stated they have corrected it but not really corrected. The authors have all the RESPONSIBILITIES for checking each and every words in the manuscript correct rather than submit the revised version fast.

7. PLOS authors have the option to publish the peer review history of their article (what does this mean?). If published, this will include your full peer review and any attached files.

Reviewer #1: No

---

## [Author Response · Author response to Decision Letter 2]

23 Aug 2020

RESPONSES TO ACADEMIC REVIEWER

We are grateful that the editors have given us another opportunity to correct and improve our manuscript for publication.

Table 3. We have rewritten Table 3. We have specified the referent category for each variable for clarity.

Typos in the Tables still noted by the academic reviewer:

- * for race was actually for gestational diabetes. An explanatory footnote has been therefore inserted under the Table 2.

- Typos have been corrected under Table 1:

o Birthweight range corrected

- Typos under Table 2:

o Maternal Age

o Gestational diabetes

o Gestational Age

NB: We also noted on rereading the manuscript that one reference was duplicated (reference 21 and 31). We therefore deleted reference 31 and the text references were therefore also updated carefully.

We hope that we have answered your queries to your satisfaction but please do not hesitate to let us know if further improvements are warranted.

Philip Kum-Nji, MD, MPH

---

## [Editor Report · Decision Letter 3]

3 Sep 2020

Group B streptococcal colonization: prevalence and impact of smoking in women delivering term or near term neonates in a large tertiary care hospital in the southern United States

PONE-D-20-15744R3

Dear Dr. Kum-Nji,

We’re pleased to inform you that your manuscript has been judged scientifically suitable for publication and will be formally accepted for publication once it meets all outstanding technical requirements.

Kind regards,

Frank T. Spradley

Academic Editor

PLOS ONE

---

## [Editor Report · Acceptance letter]

7 Sep 2020

PONE-D-20-15744R3 

Group B streptococcal colonization: prevalence and impact of smoking in women delivering term or near term neonates in a large tertiary care hospital in the southern United States 

Dear Dr. Kum-Nji:

I'm pleased to inform you that your manuscript has been deemed suitable for publication in PLOS ONE. Congratulations! Your manuscript is now with our production department. 

Kind regards, 

on behalf of

Dr. Frank T. Spradley 

Academic Editor

PLOS ONE